# Linking large-scale weather patterns to observed and modelled turbine hub-height winds offshore U.S. West Coast

Ye Liu[1], Timothy W. Juliano[2], Raghavendra Krishnamurthy[1], Brian Gaudet[1], and Jungmin Lee[3]

[1]Pacific Northwest National Laboratory, Richland, WA, 99352, USA.
[2]U.S. National Science Foundation National Center for Atmospheric Research, Boulder, CO, 80307, USA.
[3]Lawrence Livermore National Laboratory, Livermore, CA, 94551, USA.

*Correspondence to*: Ye Liu (ye.liu@pnnl.gov)

Submitted to Wind Energy Science.

**Abstract**

The U.S. West Coast holds great potential for wind power generation, although its potential varies due to the complex coastal climate. Characterizing and modelling turbine hub-height winds under different weather conditions are vital for wind resource assessment and management. This study uses a two-staged machine learning algorithm to identify five large-scale meteorological patterns (LSMPs): post-trough, post-ridge, pre-ridge, pre-trough, and California-high. The LSMPs are linked
to offshore wind patterns, specifically at lidar buoy locations within lease areas for future wind farm development off Humboldt and Morro Bay. While each LSMP is associated with characteristic large-scale atmospheric conditions and corresponding differences in wind direction, diurnal variation, and jet features at the two lidar sites, substantial variability in wind speeds can still occur within each LSMP. Wind speeds at Humboldt increase during the post-trough, pre-ridge, and California-high LSMPs and decrease during the remaining LSMPs. Morro Bay has smaller responses in mean speeds, showing increased wind speed
during the post-trough and California-high LSMPs. Besides the LSMPs, local factors, including the land-sea thermal contrast and topography, also modify mean winds and diurnal variation. The High-Resolution Rapid Refresh model analysis does a good job of capturing the mean and variation at Humboldt but produces large biases at Morro Bay, particularly during the pre-ridge and California-high LSMPs. The findings are anticipated to guide the selection of cases for studying the influence of specific large-scale and local factors on California offshore winds and to contribute to refining numerical weather prediction
models, thereby enhancing the efficiency and reliability of offshore wind energy production.

## 1 Introduction

With over 6000 megawatts (MW) of potential offshore wind-generating capacity in the development and operational pipeline, the U.S. West Coast is next in line following the successes of offshore wind deployment along the U.S. Atlantic Coast (Musial et al., 2023). The growth of wind power generation increases the dependence of the power system on variable weather and

climate (Meenal et al., 2022). As wind energy sources are highly intermittent and variable, accurate weather forecasts are essential to mitigate the related uncertainties (Frías-Paredes et al., 2017), improving decision-making and reducing cost (Turner et al., 2022; Jeon et al., 2022).

As of October 2023, five wind energy lease areas were established off the California coast — two off Humboldt County and three off Morro Bay (BOEM, 2023). Observational datasets are ideal for assessing and characterizing the wind resource. The U.S. Department of Energy funded the installation of two research buoys in these areas, equipped with lidar and other instruments to collect wind measurements for resource assessment and model evaluation (Krishnamurthy et al., 2023). However, due to the challenges associated with deploying and maintaining offshore equipment, these measurements remain limited.

The wind energy sector has greatly benefited from the use of numerical weather prediction. The High-Resolution Rapid Refresh (HRRR) model, operational at NOAA/NCEP since 2014, is a convection-permitting implementation of the Advanced Research version of the Weather Research and Forecasting (WRF-ARW) model with hourly data assimilation (Dowell et al., 2022). The 2023 National Offshore Wind dataset (NOW-23) is the latest wind resource dataset for offshore area region in the U.S., lunched by the National Renewable Energy Laboratory (NREL) (Bodini et al., 2024). The NOW-23 dataset delivers an updated and cutting-edge product to stakeholders.

The HRRR model has been proven to provide skillful forecasts of near-surface winds, leading to potential cost savings of \$14.3–\$46.6 million yearly when more advanced model configurations were applied (Fovell and Gallagher, 2022; Turner et al., 2022; Jeon et al., 2022). Despite its overall promise, the HRRR model's capabilities vary across locations and under different weather conditions (e.g., Pichugina et al., 2020). Liu et al (2024) found that HRRR tends to overestimate the turbine hub-height wind speed over complex terrain in the southeastern U.S., while Pichugina et al. (2019) reported that the HRRR model underestimated strong winds speeds ($>12$ m s$^{-1}$). Most of the evaluations focused on onshore wind energy applications due to general lack of high-quality wind profile measurements offshore (Banta et al., 2013; Myers et al., 2024; Shaw et al., 2019; Wilczak et al., 2019). Banta et al. (2017) evaluated HRRR model wind forecasts against offshore Doppler lidar measurements along the U.S. Atlantic Coast. They found an average of 1.5 to 2 m s$^{-1}$ model errors at 100 to 500 m above mean sea level (MSL).

Both onshore and offshore evaluation suggest that mean wind speed and the model bias are sensitive to weather conditions (Bianco et al., 2019; James et al., 2017, 2018). The wind speed and/or model biases tend to be larger in winter than summer in the contiguous U.S., Pacific Northwest, Great Plains, southeastern U.S., and the U.S. Atlantic offshore (Berg et al., 2021; James et al., 2018, 2017); however, an opposite trend is observed along the California offshore coast (Liu et al., 2024; Krishnamurthy et al., 2023). In addition to large-scale weather patterns, offshore wind profiles and energy production are influenced by local factors such as frontal passages, sea breezes, and low-level jets (LLJs) (Kalverla et al., 2019; Liu et al., 2024; Sheridan et al., 2024; Gaudet et al., 2022). Specifically, sea-breeze circulations entail diurnal variations in wind speed due to thermal contrast between land and sea (e.g., Gilliam et al., 2004; Burk and Thompson, 1996). During summer, this thermal contrast can cause diurnal variations in wind speed via thermal wind effect, without significant changes in wind

direction (e.g., Liu et al., 2024). The presence of the North Pacific High (NPH) system and the interaction with thermal wind

forcing, shallow marine boundary layer (MBL), and local topography often leads to a maximum wind speed at the top of the MBL (near turbine height), resulting in the formation of LLJs (Burk and Thompson, 1996).

While local factors can have a pronounced impact on near-surface winds, model biases during a period characterized by multiple weather conditions can mask local factors and ultimately lead to overlooking their impacts (Ohba et al., 2016; Liu et al., 2022; Spassiani and Mason, 2021). For instance, the approaching of a large-scale trough and ridge induces respective

southerly and northerly winds. Averaging over these two periods cancels individual effects. In this study, first, we use a two-stage clustering method to identify the predominant large-scale meteorological patterns (LSMPs) influencing the California offshore environment. Then, we characterize the wind resources under each LSMP before evaluating the HRRR model's simulated winds under these LSMPs.

## 2 Data and method

### 2.1 Lidar buoy data

The U.S. Department of Energy, in collaboration with the Bureau of Ocean Energy Management, placed two buoys equipped with Doppler lidars along the California coastline to directly observe the offshore wind resource. These buoys were positioned in the wind energy lease areas off the coasts of Humboldt and Morro Bay (Krishnamurthy et al., 2023). Over a full year, the buoys gathered data on wind patterns and turbulence from 40 to 240 m above MSL, surface meteorology, sea surface

temperature, solar radiation, two-dimensional wave spectra, and ocean current profiles including speed and direction.

The lidar on the Humboldt buoy temporarily required servicing due to a power system failure, and as a result, its observations are only available from October to December 2020 and from June to December 2021. In contrast, the Morro Bay buoys operated consistently throughout these periods (i.e., from October 2020 to December 2021). Any impact on the accuracy of lidar measurements caused by precipitation events and foggy conditions remains a subject of ongoing research. Upon

detailed examination of the carrier to noise ratio and horizontal wind speed depicted in the time-height plots during the analyzed periods, no consistent issues with the observations were identified.

### 2.2 HRRR analysis

The HRRR dataset, which is accessible through Amazon Web Services, is available in both hybrid and pressure vertical coordinates with a 3-km horizontal grid spacing (https://registry.opendata.aws/noaa-hrrr-pds/). We obtain the wind speed and

direction from HRRR at 80 m above MSL from October 2020 to December 2021. The hourly wind components are interpolated horizontally, aligning with the locations of observation sites.

## 2.3 Meteorological variables describing weather patterns

Atmospheric patterns over the North Pacific influence coastal winds, but the HRRR model's western boundary is too close to the coastline to effectively capture these circulations. To address this, the European Centre for Medium-Range Weather Forecasting 5th Generation Reanalysis (ERA5, Hersbach et al., 2020), with its global coverage, is used to perform the weather pattern classification. The hourly variables for the period of 2019-2022 at horizontal resolution of 0.25°×0.25° are obtained from https://www.ecmwf.int/en/forecasts/dataset/ecmwf-reanalysis-v5 (last accessed January 2024). The 500-hPa geopotential height ($Z_{500}$) and surface pressure ($P_{sfc}$) are obtained to describe the large-scale pressure gradient, and the 2-m temperature ($T_2$) is used for land-sea thermal contrast.

## 2.4 Large-scale meteorological weather pattern clustering

The two-stage clustering method consists of a self-organizing maps (SOMs, Vesanto and Alhoniemi, 2000; Kohonen, 1982) analysis to reduce the dimension of the input vectors ($Z_{500}$, $P_{sfc}$, and $T_2$), and a K-means cluster to further group the SOM prototypes into fewer LSMPs (Liu et al., 2023). In the first stage, we train a SOM to generate a low-dimensional discretized representation of the data in the original feature space while preserving the topological properties (relative position between the SOM nodes) of the data. In the second stage, we use the SOM prototypes as input to train the K-means method for final clustering. The SOM is a widely used clustering analysis tool (Ohba et al., 2016; Huang et al., 2022; Liu et al., 2022) that performs a topology-preserving mapping. Directly using K-means for clustering is not recommended, as K-means is highly sensitive to the positions of the initial nodes and outliers and is not suitable for high-dimensional datasets (Mingoti and Lima, 2006; Misra et al., 2020). As a result, this two-stage procedure approach combines the strengths of both SOMs and K-means while addressing their individual shortcomings. It is important to note that the success of the clustering process heavily depends on the distinctions present in the data. While directly using SOMs in this study generally captures the LSMPs, one pattern is not represented (figure not shown). This highlights the risk of missing significant patterns and generating potentially artificial symmetric results. As a result, the two-stage approach provides a reliable clustering and is used in this study.

The numbers of prototypes and LSMPs are prescribed depending on the scale of meteorological patterns. We choose a large map size, 10×10 SOM prototypes, which is sufficient to represent all possible mesoscale variations (on the order of 100 km) such as sea breezes, squall lines, and mesoscale convective complexes. Then, the silhouette score (SS) is used to determine the number of LSMPs (on the order of 1000 km) in the K-means analysis. The SS measures the separation distance between the resulting clusters: A larger SS indicates larger distinctions among the clusters (Shutaywi and Kachouie, 2021). We test 3 to 14 clusters and find that 5-cluster classification produces the largest SS (figure not shown), which therefore is chosen to perform the K-means clustering analysis. Before performing the two-stage procedure, we calculate the anomalies of input vectors by subtracting the climatological hourly mean from the timeseries at each grid point so that the annual and monthly variations are excluded. The SOM analysis is performed over 30°N–45°N, 130°W–118°W, which is chosen to include the Pacific jet exit (Athanasiadis et al., 2010).

We also compared the results of direct 5-SOM clustering with our two-stage method. The average silhouette coefficient is
0.12 for the two-stage method, compared to 0.09 for the direct 5-SOM clustering. The larger SS indicates that the resulting clusters are more well-defined and distinct.

## 2.5 Low-level jet identification

A LLJ is typically a local maximum in wind speed at altitude from near the surface up to about 2 km, yet there is no standard method for quantifying LLJs. The "fall-off" method, commonly used to determine the height of peak wind speeds (the height of the jet core), involves identifying where wind speed decreases after reaching its maximum (the jet core speed). A LLJ is recognized if the difference between the jet core speed and the next local minimum above it exceeds a certain threshold, which varies among studies (Carroll et al., 2019; Kalverla et al., 2019; Hallgren et al., 2020). Following Sheridan et al. (2024), a study that comprehensively evaluated West Coast LLJs using the same observational dataset, this study uses a 2 m s$^{-1}$ fall-off threshold to define LLJs, without specifying the vertical distance between the jet core and the threshold height as long as it is within the observational limit of 240 m above MSL. Note that due to the height limitation of 240 m, this definition will underestimate the actual number of LLJs, which will be discussed below. We only calculate LLJ from observations since HRRR provides too few near-surface points in the vertical direction for LLJ detection.

## 3 Results

Before exploring the LSMPs, we review the meteorological systems influencing near-surface winds offshore of California. The strength and location of the NPH system is the major contributor to the LSMP (Burk and Thompson, 1996). In summer, the NPH and a thermal low over the southwest U.S. create an enhanced cross-coastline pressure gradient, which primarily drives the prevailing northerly winds offshore (Brewer et al., 2012). The subsidence within the NPH induces a pronounced air temperature inversion that is most intense and lowest near the coast, capping the MBL and limiting its vertical extent. Along the coastline, a northerly to northwesterly LLJ is typically observed at the top of the MBL, resulting from the thermal wind due to significant coastal baroclinity superimposed on the generally northerly flow (Burk and Thompson, 1996). During the day, the land is typically warmer than the ocean. Above the MBL, the thermal wind is southerly aloft, and so the geostrophic winds become more northerly closer to the surface, increasing northerly wind speed until surface friction slows it down within the MBL, at around 500 m (e.g., Liu et al., 2024). In contrast, during winter, the NPH weakens, leading to distinct synoptic-scale weather conditions including storms and fronts originating from the Gulf of Alaska. The propagation of these systems can drive strong winds and a wind direction shift from northwest to southeast. The MBL also deepens during this season, influenced by the changing dynamics of the NPH.

### 3.1 SOM prototypes

In the first stage clustering, 10×10 SOM prototypes resemble the large-scale circulation modified by mesoscale perturbations (Fig. 1). Viewing Fig. 1 from left to right, the progression shows a 500-hPa high moves from west to east, coinciding with highs and lows generally rotating clockwise in the upper half of the SOMs and counter-clockwise in the lower half. From top to bottom, a 500-hPa high moves from north to south, with systems rotating counter-clockwise in the left half of SOMs and clockwise in the right half. This reflects the typical pattern evolution seen in large-scale systems though localized variation can occur. Changes in the mid-level atmosphere correspond to surface alterations. The $P_{sfc}$ generally resembles the similar patterns of $Z_{500}$, leading to a range of $P_{sfc}$ gradients that drive variable surface winds (figure not shown). Mid-level high pressure, often associated with downdrafts, inhibits cloud formation, thereby increasing surface solar radiation and temperature (Dadashazar et al., 2020). Conversely, mid-level low pressure can induce opposite changes. Meanwhile, land exhibits larger temperature variations than ocean due to its smaller heat capacity. As a result, positive and negative $T_2$ anomalies are found over the land area underlying respective positive and negative $Z_{500}$ anomalies.

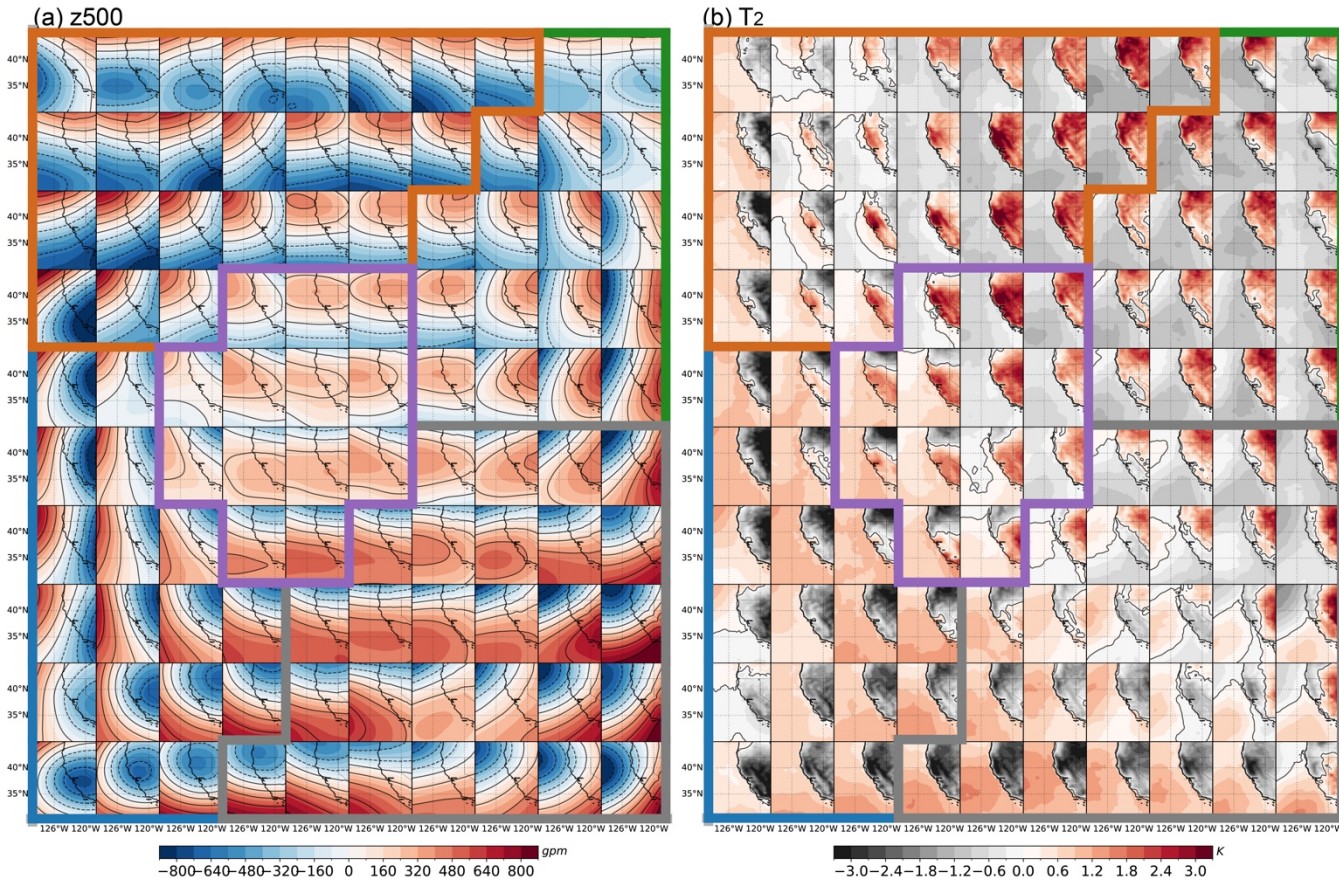

Figure 1: Composite anomalies of $Z_{500}$ and $T_2$ for each SOM prototype. The red shading and solid contours indicate positive anomalies, and blue and dashed contours indicate negative anomalies. The coloured lines outline the five LSMPs: post-trough (blue), post-ridge (green), pre-ridge (orange), pre-trough (grey), and California-high (purple).

## 3.2 Dominant large-scale meteorological weather patterns and associated wind patterns

In the second stage clustering, the K-mean analysis produces five LSMPs. The first two LSMPs, each accounting for 17% of the total hours during the study period, resemble large-scale ridges and troughs centred over the western U.S., with the buoys located behind (west) of those systems (Fig. 2a, b). For the post-trough weather pattern, the passing of the mid-level trough and the following ridge of high pressure intensifies the cross-coastline pressure gradient, which enhances the northerly winds offshore. Although the cold northerly winds cool the land surface, the land-sea thermal contrast still exists and further accelerates the northerly winds through the thermal wind effect. Consequently, the post-trough LSMP drives strong offshore winds along the coast, with enhanced expansion fans (an area of high wind speed) downwind of Cape Mendocino and Point Conception (Fig. 3a).

The second LSMP, namely post-ridge, is associated with blocking systems over the western U.S.; this is an elongated area of relatively high atmospheric pressure. The high-pressure systems are associated with subsiding air, which discourages cloud formation and leads to stable weather conditions. The inland surface temperature can be higher due to the increased solar radiation and the general downward motion of air, which warms adiabatically as it descends. Under post-ridge conditions, the weakened cross-coastline pressure gradient decreases the prevailing surface northerly wind (Fig. 3b). In contrast, the enhanced land-sea thermal contrast along the Oregon coast tends to increase offshore winds through the thermal wind theory (Liu et al., 2024), which mitigates the overall decrease in offshore wind speed.

The third and fourth LSMPs resemble large-scale ridges and troughs centred over the Pacific Ocean, and the buoys are located ahead (east) of those systems (Fig. 2c, d), accounting for 26% and 28% of the total hours, respectively. Like the post-ridge pattern, the pre-ridge pattern is associated with high-pressure systems favourable to subsidence, warm land surface, and decreased offshore winds (Fig. 3c). In contrast, the pre-ridge pattern features an anomalous north-to-south pressure gradient at 500 hPa and a strong surface pressure gradient centred offshore of northern California and Oregon, accelerating wind speed. Meanwhile, the warmer land surface over the northern California mountains further increases the offshore winds. As a result, the pre-ridge increases (decreases) wind speed off northern (southern) California.

For the fourth LSMP, namely pre-trough, the buoys are located ahead (east) of a mid-level trough, where the dynamics can create conditions favourable for a cold front and strong convection. The enhanced NPH over California and the Aleutian Low intensify the pressure gradient at 500 hPa and the surface, which often is associated with a cold front approaching California from the northeast Pacific Ocean. As a cold front approaches, the tightened pressure gradient can lead to an increase in wind speeds and change the wind direction to south or southwest (Fig. 3d). This occurs more frequently offshore of northern California and thus has less impact on the wind speed in the south.

The fifth LSMP, namely California-high, accounts for 11.4% of the hours and exhibits an anomalous high at 500 hPa centred offshore of California (Fig. 2e). The mid-level and surface pressure and temperature patterns are similar to the pre-ridge LSMP, except they have larger magnitudes. Therefore, the area with accelerated winds extends from northern California to the south (Fig. 3e).

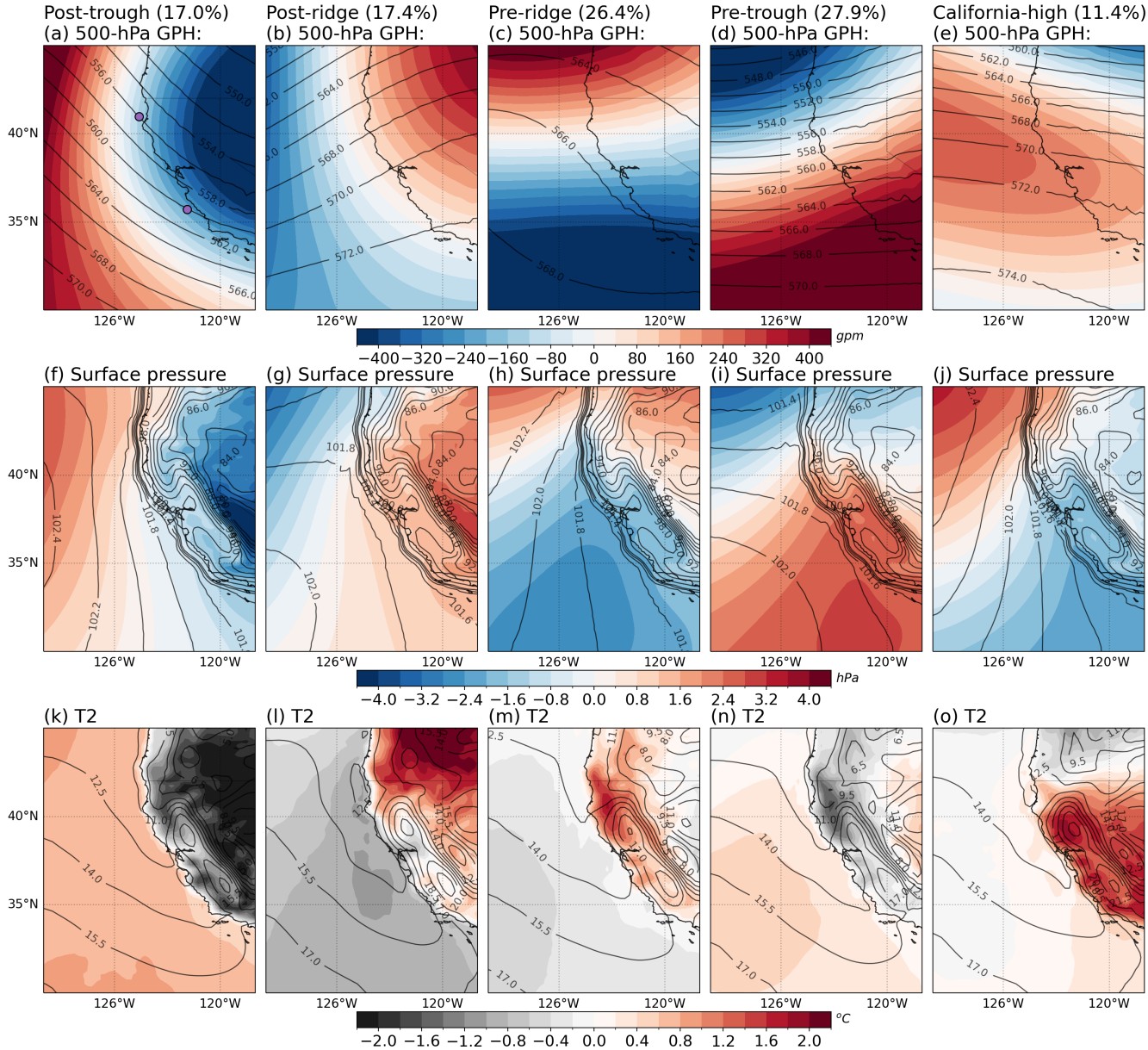

**Figure 2: Spatial distributions of mean (contour) and anomalies (shading) of 500 hPa GPH (a-e), surface pressure (f-j), and T2m (k-o). The value at the top of each column indicates the frequency of each LSMP. The purple dots on panel (a) indicate the lidar buoy locations at Humboldt in the north and Morro Bay in the south.**

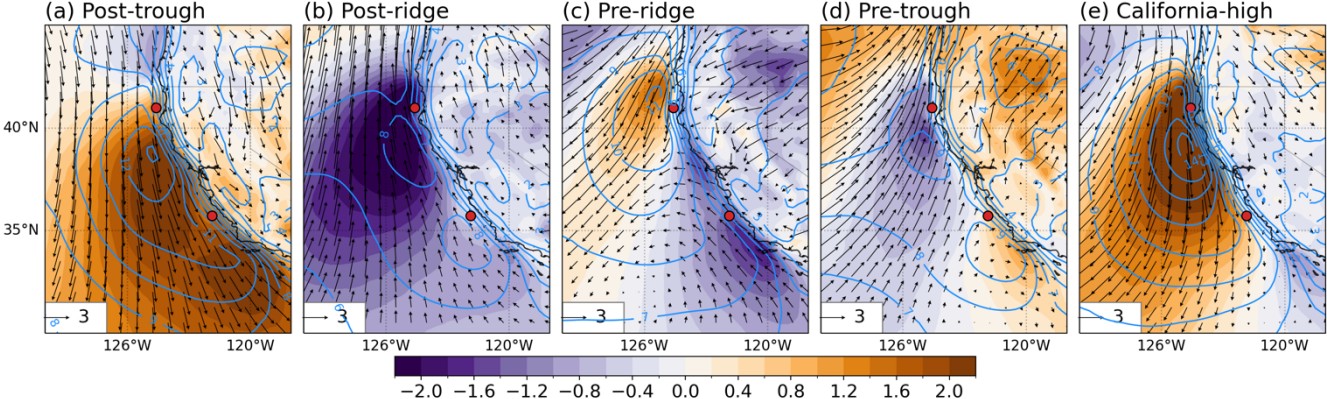

**Figure 3: Spatial distributions of mean (contour) and anomalies (shading and vector) of 80-m wind speed and direction from HRRR. The red dots on panel (a) indicate the lidar buoy locations at Humboldt in the north and Morro Bay in the south.**

210

**Table 1: Summary of LSMPs and the associated wind patterns**

| LSMP | Frequency (%) | Weather pattern | Wind pattern |
|---|---|---|---|
| Post-trough | 17.0 | 500-hPa trough centred over western U.S land. Intensified cross-coastline surface pressure gradient. Weakened land-sea thermal contrast. | Strong northerly to northwesterly winds with expansion fans downstream the capes. |
| Post-ridge | 17.4 | 500-hPa ridge centred over western U.S. land. Weakened cross-coastline surface pressure gradient. Enhanced land-sea thermal contrast. | Decreased overall wind speeds offshore of north California. |
| Pre-ridge | 26.4 | 500-hPa ridge centred over north Pacific Ocean. Anomalous north-to-south surface pressure gradient, with enhanced cross-coastline surface pressure gradient offshore of north California, and decreased pressure gradient in the south. Enhanced land-sea thermal contrast. | Increased northerlies offshore of north California, and decreased wind speed in the south. |
| Pre-trough | 27.9 | 500-hPa trough centred over north Pacific Ocean. Anomalous south-to-north surface pressure gradient, with decreased cross-coastline surface pressure gradient offshore of north California, and decreased pressure gradient in the north. Decreased land-sea thermal contrast. | Reduced northerlies offshore of north California, and increased wind speed in the south. |
| California-high | 11.4 | 500-hPa high centred over California. Enhanced cross-coastline surface pressure gradient. Increased land-sea thermal contrast. | Strong northerly winds offshore of central and north California. |

## 3.3 80-m wind regimes at buoy locations

Wind patterns at Humboldt and Morro Bay exhibit distinct responses to LSMPs and smaller-scale atmospheric disturbances as recognized by the SOM prototypes. Figure 4 presents the average 80-m wind speed for each SOM prototype. Post-trough

conditions reveal stronger northerly to northwesterly winds along the coast. The Humboldt buoy, situated at the boundary of the northern expansion fan, experiences a pronounced horizontal wind speed gradient (Fig. 3a), leading to a wide range of mean wind speeds from 1 to 14 m s$^{-1}$. In contrast, the Morro Bay buoy consistently records high wind speeds.

Under post-ridge conditions, anomalous southerlies are most intense offshore of northern California, significantly reducing wind speeds at Humboldt and potentially causing a wind direction change when the southerly anomaly is greater than the prevailing north wind. It is interesting to note that wind speeds observed in certain SOM prototypes, such as the high values in the top-right cluster for Humboldt, result from the interaction between the prevailing wind direction and the anomalies induced by the LSMP. Typically, the prevailing winds in this region are northerly, while the LSMP tends to induce a southerly wind anomaly. In most cases, this anomaly reduces wind speed by counteracting the northerly flow. However, when the southerly anomaly is strong enough, it can either shift the wind direction to the south or surpass the strength of the prevailing northerly winds, leading to an increase in wind speed. As such, when the mid-level ridge intensifies, these offshore winds can become southerlies with speeds exceeding 15 m s$^{-1}$ in extreme cases. At Morro Bay, wind speeds generally weaken to 6–11 m s$^{-1}$.

The pre-ridge LSMP results in increased mean wind speeds off northern California and decreased speeds to the south. This wind pattern is manifested by anomalies in surface temperature. The dominance of NPH-induced clear skies leads to warmer land temperatures that enhance the land-sea thermal contrast, thereby accelerating offshore winds. Across the SOM prototypes, the wind speed at Humboldt range of 6–15 m s$^{-1}$, while the Morro Bay buoy records a range of 4–10 m s$^{-1}$.

Pre-trough conditions, marked by an approaching mid-level trough, generate moderate to high winds (7–15 m s$^{-1}$) at Humboldt, with wind directions shifting from northwest to southwest during surface frontal passages. The impact of the mid-level trough on the Morro Bay buoy is minimal, with prevailing northwesterlies under the influence of the NPH. However, if the trough deepens, wind speeds at Morro Bay may transition from weaker northwesterlies to stronger southwesterlies.

During California-high conditions, both Humboldt and Morro Bay record increased wind speeds due to an enhanced NPH and the amplified land-sea thermal contrast. The mean wind speed exceeds 12 m s$^{-1}$ at Humboldt and 10 m s$^{-1}$ at Morro Bay.

It is important to note that the classification of LSMPs does not imply that each LSMP is associated with a narrowly defined wind speed. Rather, each LSMP reflects a dominant synoptic environment under which the prevailing direction and magnitude of offshore winds are modulated. Within any single LSMP, a range of mesoscale and local factors (e.g., frontal passages, varying thermal contrasts, topographic influences, and boundary layer structures) can lead to substantial variability in observed wind speeds. For instance, under post-ridge conditions, most SOM prototypes show a decrease in wind speed due to the induced southerly anomaly; however, a few prototypes exhibit unexpectedly high wind speeds when the southerly anomaly surpasses the prevailing northerly flow. This illustrates that while the LSMP framework provides a useful synoptic-scale context, it is primarily a classification tool rather than a deterministic method, and thus cannot eliminate the inherent complexity and spread in the local wind speed distributions.

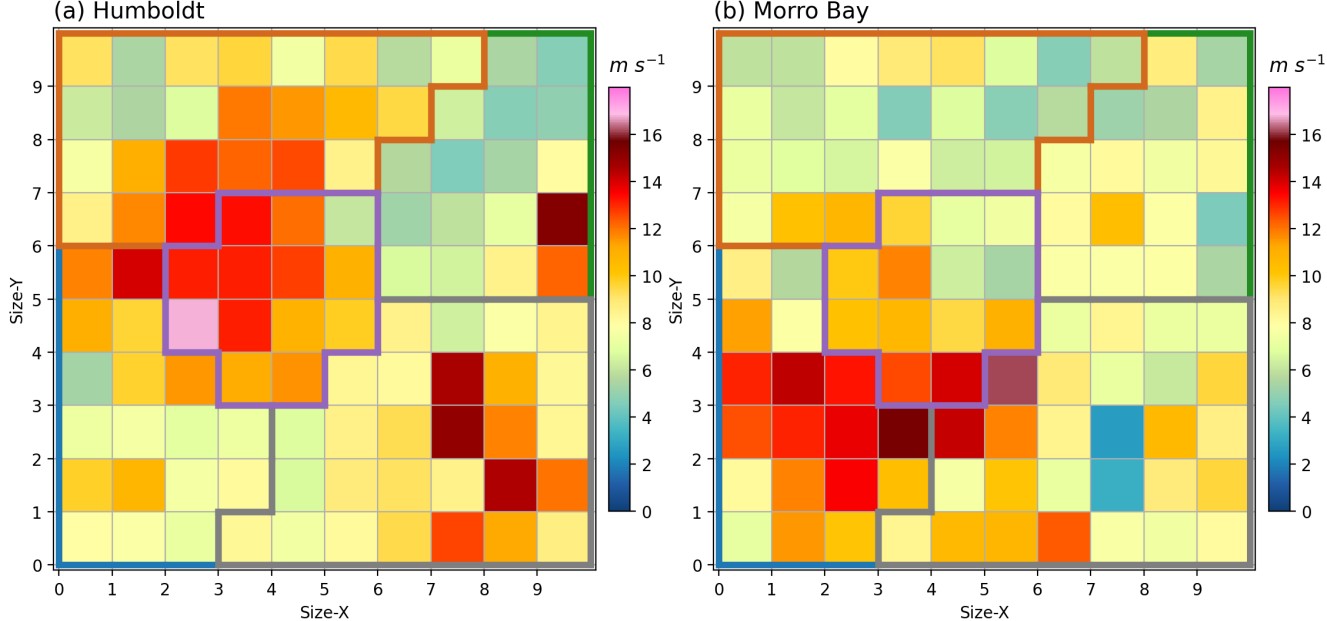

Figure 4: Mean 80-m wind speed of each SOM prototype at Humboldt (a) and Morro Bay (b). The coloured lines outlined the five LSMPs: post-trough (blue), post-ridge (green), pre-ridge (orange), pre-trough (grey), and California-high (purple).

250  ## 3.4 Observed and modelled diurnal variation

The diurnal variation in surface heating alters the wind speed and direction via the thermal wind effect. The thermal wind effect results in northerly winds relative to the free troposphere during the day and southerly winds at night. The maximum speed occurs a few hours after the peak of baroclinity in the mid-afternoon as the inertial turning of the land-sea breeze circulation to the free troposphere wind direction through the thermal wind effect (Burk and Thompson, 1996).

255  At Humboldt, the prevailing free tropospheric wind direction varies across the LSMPs, showing a northerly trend during the post-trough, pre-ridge, and California-high conditions and a southwesterly shift during the post-ridge conditions (Fig. 5). The maximum speeds are observed near midnight (09–12 UTC) under northerly free troposphere wind conditions and in the afternoon (00 UTC) during southwesterly conditions. Large-scale cold fronts primarily influence the winds during pre-trough conditions, resulting in small diurnal variations. In contrast, at Morro Bay, the maximum speed is observed from evening to

260  midnight (00–06 UTC) due to the prevailing northerly to northwesterly winds throughout all LSMPs.

The diurnal variations in 80-m wind speed modelled by HRRR are compared with observations (Fig. 5). HRRR does a good job of capturing the mean wind speed at Humboldt, with biases ranging from -0.5 to 0.1 m s$^{-1}$. The HRRR effectively captures diurnal variations during post-trough and post-ridge LSMPs but underestimates the variation during pre-ridge conditions. At Morro Bay, the HRRR produces small biases during post-trough, post-ridge, and pre-trough conditions, while

265  it largely overestimates the daily mean during the pre-ridge LSMP by 2.2 m s$^{-1}$ and California-high LSMP by 2.6 m s$^{-1}$. The overestimation may be connected to frequent LLJs occurring during these two LSMPs, which results in a large vertical wind

speed gradient. Liu et al. (2024) also reported that an overpredicted land-sea thermal contrast can lead to an overestimation of wind speed. Generally, HRRR tends to reproduce the diurnal variations with a slight delay during most LSMPs except for pre-trough, where the model shows no diurnal changes.

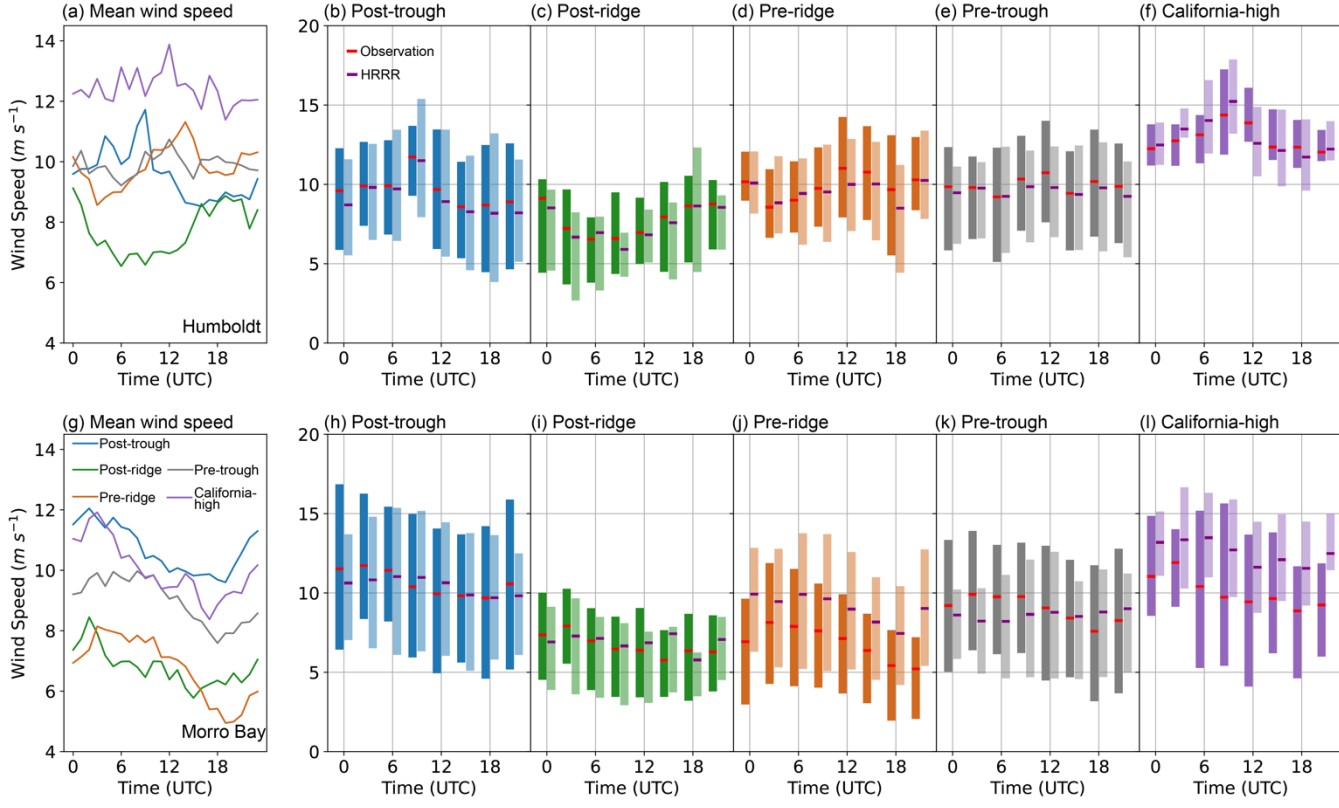

**Figure 5: Diurnal variations of wind speed at Humboldt (a-f) and Morro Bay (g-l). Observed mean wind speed associated with each LSMP (a, g) and box plot comparison for individual LSMPs (b-f) and (h-l). The line in the centre of each bar indicates the mean value and the limits of the bars indicate the first quartile (Q1) to the third quartile (Q3) of the data.**

### 3.5 Observed low-level jet

The LLJ offshore of California is often characterized by a strong vertical wind speed gradient, which can introduce significant biases in modelled wind speed at 80 m (Liu et al., 2024). We examine the occurrence, jet-core height, and jet-core wind speed across various LSMPs (Fig. 6). Out of all the lidar observations, 1107 (4%) and 1911 (4%) record LLJ occurrences at Humboldt and Morro Bay, respectively, consistent with the findings by Sheridan et al. (2024) using the same lidar dataset. At Humboldt, the pre-ridge LSMP has the highest LLJ occurrence (406 LLJs), with the occurrence of jet-core height peaking at 160 m. The pre-ridge and pre-trough LSMPs record 309 and 238 LLJs, respectively, with a bimodal distribution of occurrence of jet-core height peaking near 140 and 220 m for both LSMPs. Interestingly, the post-trough and California-high LSMPs record few LLJs despite being associated with high-speed winds. This is likely because high-speed winds consistently appear throughout the lidar measuring range of 40–240 m without forming a jet structure.

The mean jet-core wind speed at Humboldt varies between 9.7 and 11.4 m s$^{-1}$ across all LSMPs and generally increases to its height with a maximum at 200 m (figure not shown). Larger variations appear associated with individual LSMPs. The California-high LSMP has the largest mean jet-core wind speed of 18.3 m s$^{-1}$ at 220 m. For other LSMPs, the maximum mean jet-core speed ranges from 13.9 to 16.5 m s$^{-1}$ at various heights.

Similar jet characteristics are observed at Morro Bay during most LSMPs, except for the California-high, which records the most LLJ occurrences (663 LLJs) at Morro Bay. The discrepancy between the two locations during the California-high LSMP may be connected to the typically lower MBL at Morro Bay compared to Humboldt (Zhou et al., 2020), resulting in lower jet cores peaking at 160 m, within the lidar range. The maximum mean jet-core wind speed varies from 12.5 to 17.4 m s$^{-1}$. Despite the opposite sign anomalous 100-m wind speed at two locations during the pre-ridge and pre-trough LSMPs (Fig. 3c, d), minor differences are observed in jet characteristics. This supports previous literature reporting that the LLJ is a mesoscale phenomenon modified by local meteorology and topography.

It is important to note that the lidar's maximum measurement height limitation (220 m) likely results in an underrepresentation of LLJ occurrence at heights above 220 m. The consistent increase in mean wind speed with height suggests potential jet cores above the highest measurement. To the best of our knowledge, long-term LLJ measurements do not exist in this region. Therefore, it is difficult to know the true frequency of LLJ conditions. Nonetheless, we anticipate that the limited range of the lidar contributes to an underestimation of LLJ frequency, which reanalysis and global climate models estimate to be ~20-30% annually off the California coastline (e.g., Lima et al., 2022, Juliano et al., 2024).

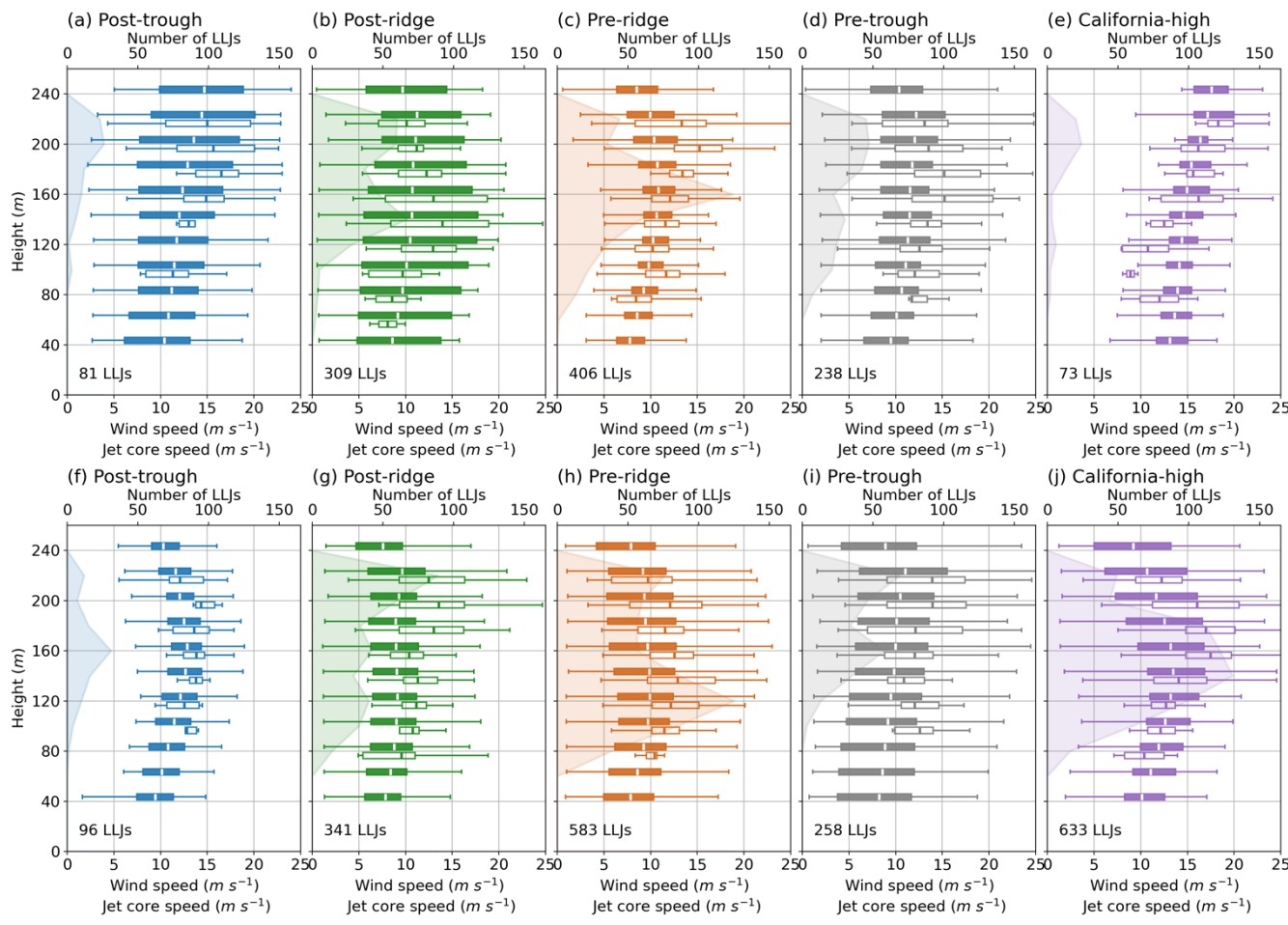

**Figure 6:** Observed wind speed profile (filled box), occurrence of jet-core height (shading), and jet-core wind speed (open box) at Humboldt (a-e) and Morro Bay (f-j) for each LSMP. The value in the lower-left indicates the number of LLJ events observed during each LSMP. The box extends from the first quartile (Q1) to the third quartile (Q3) of the data, with a dash line at the mean. The whiskers extend from Q1-1.5× (Q3-Q1) to Q3+1.5×(Q3-Q1).

## 4 Conclusion

In this study, we use a two-stage clustering method to identify the LSMPs influencing near-surface winds off the California coast. The 10×10 SOM prototypes resemble the evolution of weather systems such as high-pressure systems moving eastward and southward (Fig. 1), driving variations in wind speed and direction at the Humboldt and Morro Bay buoys. The SOM prototypes are aggregated using the K-means method into five LSMPs: post-trough, post-ridge, pre-ridge, pre-trough, and California-high (Fig. 2). The post-trough and post-ridge LSMPs resemble west-to-east mid-level and surface anomalies, enhancing and reducing the cross-coastline pressure gradients, respectively, leading to accelerated and deaccelerated offshore winds (Fig. 3). The pre-ridge and pre-trough LSMPs resemble north-to-south mid-level and surface anomalies with intensified anomalous surface pressure gradient offshore of north California and Oregon. The pre-ridge LSMP increases wind speed at

Humboldt and slightly decreases wind speed at Morro Bay. Opposite changes are observed in mean wind speed at the two locations associated with pre-trough LSMP. However, a strong surface front can drive relatively strong southwesterly winds that are larger in magnitude than the prevailing northerly winds. Nonetheless, these results should not be interpreted to mean that each LSMP strictly enforces a single wind speed regime. A wide range of wind speed outcomes can occur, influenced by local and mesoscale processes.

The offshore near-surface winds are modified by the diurnally varying land-sea thermal contrast (Fig. 5). The maximum wind speeds occur a few hours after peak baroclinity in the afternoon and vary with the prevailing free tropospheric wind direction, which is influenced by different LSMPs. The HRRR model generally captures these variations well at Humboldt, with minor biases, but it overestimates mean wind speeds at Morro Bay during pre-ridge and California-high LSMPs, possibly due to frequent LLJ occurrences and an overpredicted land-sea thermal contrast. HRRR also tends to show a slight delay in reproducing diurnal variations and does not reflect changes during pre-trough conditions.

At Humboldt, the highest LLJ occurrence is during the pre-ridge LSMP, with jet-core heights peaking at 160 m, while at Morro Bay, the California-high LSMP records the highest number of LLJs (Fig. 6). The mean jet-core wind speed at Humboldt ranges from 9.7 to 11.4 m s$^{-1}$, increasing with height, and the California-high LSMP shows the highest mean speeds up to 18.3 m s$^{-1}$. Despite some differences in wind speeds and LLJ characteristics between Humboldt and Morro Bay, the general consistency between the two locations (800 km apart) suggests the LLJ is a meso-alpha scale phenomenon modified by local conditions.

The identified model biases have significant implications for wind farm development, particularly in offshore environments where accurate wind resource assessments are essential. For instance, the overestimation of wind speeds in certain LSMPs, such as pre-ridge and California-high conditions, could result in overestimating potential energy output. To address this, data users should approach HRRR model outputs cautiously under these conditions and incorporate model uncertainties into their assessments. Beyond the mean status of wind speed, future studies could link the wind power features like ramp frequency and intensity to LSMPs. Practical measures, such as utilizing ensemble forecasts or combining multiple models, can help mitigate the effects of these biases on wind farm siting and design decisions.

This study introduces a new approach to characterizing offshore winds and associated model biases, linking them to LSMPs. The methodology used for evaluating HRRR performance under different LSMPs can be applied to other numerical weather prediction models. This approach not only identifies model strengths and weaknesses but also provides valuable insights into how environmental factors influence airflow, aiding predictive studies. By connecting model performance to LSMPs, this method promotes mechanism analysis, fostering studies on a deeper understanding of the physical processes behind wind patterns. Furthermore, the results are anticipated to guide the selection of cases for studying the influence of specific large-scale and local factors on winds off the California coast, which will aid in refining numerical weather prediction models, thereby enhancing the efficiency and reliability of offshore wind energy production.

In addition to the LSMP-based classification used in this study, there is potential for an alternative approach that clusters directly on 80-m wind speeds before identifying the corresponding large-scale meteorological patterns. This reverse

classification method could better capture the variability in wind speeds that is particularly relevant for practical applications, such as wind farm development. By focusing on the wind resource itself, this approach may provide improved insights into local wind speed patterns and reduce the occurrence of outliers within clusters. Our team is actively exploring this method to complement the current LSMP-based analysis and further refine wind resource assessment techniques.

## Code and Data availability

The lidar buoy data utilized in this study are publicly available from the U.S. Department of Energy for Humboldt (https://doi.org/10.21947/1783809) and Morro Bay (https://doi.org/10.21947/1959721). ERA5 is available through the Copernicus Climate Change Service Climate Data Store at http://cds.climate.copernicus.eu. The HRRR analysis is publicly available at https://registry.opendata.aws/noaa-hrrr-pds/.

## Author contributions

YL and RK conceived the idea. YL performed the research, analysed the data, and prepared the manuscript with contributions from all co-authors. All authors contributed to the paper edits and technical review.

## Competing interests

The contact author has declared that none of the authors has any competing interests.

## Acknowledgement

This research has been supported by the U.S. Department of Energy, Office of Energy Efficiency and Renewable Energy, and Wind Energy Technologies Office. PNNL is operated for the U.S. DOE by the Battelle Memorial Institute under contract DE-A05-76RL0 1830. JL's portion of this work was prepared by LLNL under Contract DE-AC52-07NA27344. LLNL-JRNL-863111.

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
