# Peer review of "Linking large-scale weather patterns to observed and modelled turbine hub-height winds offshore U.S. West Coast"

_Wind Energy Science, 2024_

## Author Comment (AC1)

**Reviewer #1**

Review of "Linking weather patterns to observed and modelled turbine hub-height winds offshore U.S. West Coast" by Liu et al.

General comments: This manuscript provides a very interesting analysis of turbine-height wind speeds observed by two floating lidar buoys in coastal California waters. The manuscript is well-written and easy to read, and the figures are all well-composed and enlightening. My comments are mostly minor, however, two issues are more substantial and may require non-trivial revisions.

We thank the reviewer for their thoughtful and constructive comments and suggestions, which has substantially improved the quality of the manuscript. We have addressed all the reviewer's concerns and revised the manuscript accordingly. Our point-by-point responses are in blue and the modifications to the manuscript are quoted in green.

The first has to do with the statement on line 100-101, which implies that a symmetrically paired set of nodes is not independent, and that a lack of independence is an undesirable feature. But is that true? Consider for example the statement that the authors make in the introduction about model errors being different for northerly wind versus southerly winds (lines 63-64) and how it is important to treat these two separately. The statement on line 100 suggests that having a symmetric northerly and southerly pair is undesired, in contradiction to the statement on line 63. Also, does applying k-means clustering to the SOMs then remove either the northerly or southerly SOM because they are not independent?

Thanks for the great comments. We acknowledge the presence of northerly and southerly anomalies, and while symmetry is expected. However, there are cases where the paired weather phenomena differ significantly in their impact. For instance, cold and warm fronts exhibit different patterns and effects on wind speed. Similarly, certain phenomena, like atmospheric rivers, do not have a direct counterpart. Thus, although there is general symmetry, we anticipate distinctions between positive and negative pairs, and not all phenomena are paired.

Regarding the SOM classification, it holds the risk of producing nearly symmetrical patterns, which can be misleading if such symmetry is not inherent in the data. As we noted in our 2023 paper (Liu et al., 2023), this issue arose when we attempted to classify four weather regimes (Figure R1B). Although the regimes were somewhat paired, they were not exact opposites. However, the 4-node SOM analysis (Figure R1A) generated almost identical opposite patterns, e.g. regime-1 vs regime-2, which did not align with the natural variability in the data.

In this study, while some degree of symmetry is present, to avoid misclassification, we chose to apply the two-stage method. Applying k-means clustering to the SOMs doesn't remove the northerly or southerly patterns. Instead, the two-stage method separates the northerly and southerly patterns in a more natural way.

To avoid confusion, we have removed this statement on line 100-101 in the revision.

[Figure]

**Figure R1 Weather Regime (500 mb geopotential height anomaly) based in Liu et al., 2023. (A) results using 4-node SOM and (B) results using the two-stage (SOM+K-means) method. The percentage numbers indicate the occurrence of each weather regime.**

35 Although the sequential combination of SOMs and K-means clustering sounds reasonable at first glance, it is not clear what this does in practice. The manuscript would be improved if the authors provided a description of what the procedure does using some real-world meteorological examples (Northerly vs southerly flow; onshore vs offshore flow; strong winds versus weak winds; etc). I also note that I cannot find this information in the 2023 paper by Liu et al.

40 To demonstrate how the two-stage method works in clustering weather patterns based on 500-hPa geopotential height (z500), surface pressure, and 2-m temperature, we first use SOM as a dimensionality reduction technique to aggregate similar weather patterns. This step generates 10x10 SOM prototypes, which can be seen as a lower-dimensional representation of the original data. These prototypes capture the climatological propagation of weather systems. For example, using z500 (Figure R2A), each row from left to right shows the propagation of atmospheric waves moving west to east or rotation of the highs and lows. As more SOM nodes are used, finer details of the wave propagation and

45 smaller-scale features become visible. In the second stage, the K-means clustering is used to further aggregate them into large-scale patterns. As we can expect, small-scale variations will be lost when aggregated to a few nodes.

[Figure]

**Figure R2** Illustration of the two-stage method using z500 as an example. (A) 10x10 SOM maps and (B) five clusters by applying K-means clustering on SOMs.

Related to this, the combination of SOMs and K-means results in 5 LSMPs. If one only calculated 5 or 6 SOM nodes, would they give anything substantially different from these 5 LSMPs? To first order, I would expect them to be very similar. If the authors were to make this comparison, and find that the 5 or 6 SOM nodes are in fact substantially different from those from the combination procedure, then it would strongly support their contention that the two-step process is necessary. Without that test, I remain skeptical.

Thank you for your suggestion to compare the results from using 5 SOM nodes directly versus the two-step SOM/K-means method. As shown in the Figure R3, directly using 5 SOM nodes produces results similar to the first four LSMPs but fails to capture the fifth LSMP, which is present in the SOM prototypes (please see Fig R2A) generated by the two-step process in this study. This highlights a limitation of using SOM alone, where certain key patterns may not be represented. The success of the clustering process highly depends on the distribution of the data points. Directly using SOMs to classify, as demonstrated, carries the risk of missing important patterns and possible artificial symmetric patterns, making the two-step SOM/K-means approach a safer and more reliable method for ensuring all significant LSMPs are captured.

The following sentences has been added to Line 113-117.

It is important to note that the success of the clustering process heavily depends on the distinctions present in the data. While directly using SOMs in this study generally captures the LSMPs, one pattern is not represented (figure not

shown). This highlights the risk of missing significant patterns and generating potentially artificial symmetric results. As a result, the two-stage approach provides a reliable clustering and is used in this study.

[Figure]

**Figure R3 Comparison of the LSMPs identified using (A) two-step SOM/K-means approach and (B) SOM clustering.**

The second major issue has to do with Figure 4. This is a very nice figure, and very informative, and it helps to make a bit clearer the impacts of the two-step clustering process. However, I am surprised that the clusters are grouped as they are. For example, in Fig. 4a for Humboldt, the top-right K-means cluster has two very high wind speeds SOMs (red and orange squares) that are real outliers from the rest of the members. Likewise, the Morro Bay bottom right K-means group has two blue squares that are outliers. This implies that the SOM/K-means clustering method based on 500 MB geopotential, Psfc, and T2 is not always the best way to organize the data if one is interested in 80 m offshore winds. Would it make sense to run the process in reverse, and find 80 m wind speed SOMs/K-mean clusters and then find the corresponding large-scale weather patterns? Alternatively, is it possible to force the K-means clusters to have slightly modified SOM members such that these large outliers go into different K-means clusters?

Thank you for your insightful feedback regarding the clustering process shown in Figure 4. We recognize that the presence of higher wind speeds in the top-right cluster for Humboldt and the blue squares in the bottom-right cluster for Morro Bay may seem like outliers. However, we would like to clarify that these wind speeds are not outliers but rather reflect the influence of both the prevailing wind direction and the anomalies induced by the LSMPs.

Figure 4 demonstrates how wind speed is influenced by both the magnitude and direction of the prevailing wind (typically from the north in this region) and the anomalies induced by the LSMPs. Take, for instance, the top-right K-means cluster in Figure 4a (post-ridge LSMP). The LSMP in this case induces a southerly wind anomaly at Humboldt, which typically reduces wind speeds by opposing the prevailing northerly flow (as seen in Figure R4). In most scenarios, the southerly anomaly diminishes the total wind speed unless the anomaly becomes strong enough to reverse the wind direction to the south. In extreme cases, the southerly winds can even exceed the strength of the prevailing northerlies, leading to an overall increase in total wind speed. This interaction between the prevailing and anomalous

wind patterns explains the higher wind speeds observed in certain SOM clusters under the same LSMP. However, such extreme events represent only 2% of the total cases, which is why a general reduction in wind speed is observed during this LSMP.

95    The following text has been added to Line 222-227.

It is interesting to note that wind speeds observed in certain SOM prototypes, such as the high values in the top-right cluster for Humboldt, result from the interaction between the prevailing wind direction and the anomalies induced by the LSMP. Typically, the prevailing winds in this region are northerly, while the LSMP tends to induce a southerly wind anomaly. In most cases, this anomaly reduces wind speed by counteracting the northerly flow. However, when

100   the southerly anomaly is strong enough, it can either shift the wind direction to the south or surpass the strength of the prevailing northerly winds, leading to an increase in wind speed.

Your suggestion of reversing the classification process, i.e., starting with 80-m wind speeds and then finding the corresponding large-scale weather patterns, presents an interesting alternative. While this method might better capture wind speed variability, it addresses a different scientific question than our current approach, which focuses on how

105   large-scale meteorological patterns influence wind conditions. Our current LSMP-based classification helps link wind patterns to synoptic-scale processes, offering insights into atmospheric dynamics and improving model evaluation. On the other hand, a wind-speed-first classification would be more practical for applications such as wind resource assessments, and we are actively exploring this alternative approach to complement our current method. We have added the following paragraph to the end of the manuscript to highlight the wind speed-based classification approach.

110   In addition to the LSMP-based classification used in this study, there is potential for an alternative approach that clusters directly on 80-m wind speeds before identifying the corresponding large-scale meteorological patterns. This reverse classification method might better capture the variability in wind speeds that is particularly relevant for practical applications, such as wind farm development. By focusing on the wind resource itself, this approach may provide improved insights into local wind speed patterns and reduce the occurrence of outliers within clusters. Our

115   team is actively exploring this method to complement the current LSMP-based analysis and further refine wind resource assessment techniques.

[Figure]

**Figure R4 Wind speed and direction of SOMs associated with post-ridge LSMP. (A) Mean wind speed and direction and (B) anomalous wind speed and direction. The red boxes outline the two SOMs with strong winds.**

Specific comments:

Line 1: Would the title be more accurate if it said "Linking Large-Scale Weather Patterns …" since those are the only weather patterns investigated?

It has been changed in the revision.

Line 13 "resource assessment"

Corrected. Thank you.

Line 14: From symmetry, I would have expected that a "California Low" would also have been a LSMP. Why isn't it the 6th LSMP?

130 We agree that there are some instances resembling a low-pressure center over California; please see the lower left panels in Figure 1a in the main text or Figure R2a in the response. However, these patterns have fewer occurrences and resemble the post-ridge pattern. Therefore, these patterns are clustered to the post-trough pattern.

Lines 40-43. An additional offshore reference that could be added here is Myers et al. 2024: Evaluation of Hub-Height Wind Forecasts Over the New York Bight.  Wind Energy, https://doi.org/10.1002/we.2936.

135 The reference has been included in the revision. Thank you.

Line 49. An additional reference here for HRRR biases is Bianco, L et al., 2019: Impact of model improvements on 80 m wind speeds during the second Wind Forecast Improvement Project (WFIP2), Geosci. Model Dev., 12, 4803–4821, https://doi.org/10.5194/gmd-12-4803-2019.

The reference has been included in the revision. Thank you.

140 Line 54: circulations entail

Corrected.

Line 64-65: influencing the California offshore environment.

Corrected. Thank you.

Line 100: What kind of numbers are considered small here?  SOM analyses typically use 10-30 nodes, are those
145 considered to be small?

We have removed this statement in the revision.

Lines 116, 118: A LLJ

Corrected in the revision.

Lines 121-123: "this study uses a 2 m s-1 fall-off threshold to define LLJs, without specifying the vertical distance
150 between the jet core and the threshold height as long as it is within the observational limit of 240 m above MSL."  The authors should note that due to the height limitation of 240m that this definition will certainly underestimate the number of true LLJs.

The following sentence has been added to Line 136-137.

Note that due to the height limitation of 240 m, this definition will underestimate the actual number of LLJs, which
155 will be discussed below.

Line 141.  I don't always see this.  For example, in Fig.1a, in the third row from the bottom the highs and lows are definitely rotating counter-clockwise from left to right.

Good point. We have modified as follow in the revision in Line 154-159.

In the first stage clustering, 10×10 SOM prototypes resemble the large-scale circulation modified by mesoscale perturbations
160 (Fig. **Error! Reference source not found.**). Viewing Fig. **Error! Reference source not found.** from left to right, the progression shows a 500-hPa high moves from west to east, coinciding with highs and lows generally rotating clockwise in the upper half of the SOMs and counter-clockwise in the lower half. From top to bottom, a 500-hPa high moves from north to south, with systems rotating counter-clockwise in the left half of SOMs and clockwise in the right half. This reflects the typical pattern evolution seen in synoptic systems though localized variation can occur.

165 Line 204: causing a wind direction change

Corrected.

Line 235: during the pre-ridge LSMP

Corrected. Thank you.

Figure 5: Another very nice figure! The caption is a bit confusing however. Should the phrase "The line in the centre
170    of each box indicates the mean value and the extends of the box indicate the …" say "The line in the centre of each bar
indicates the mean value and the limits of the bars indicate the …?

The caption has been changed as suggested.

Line 253-254: See previous comment for lines 121-123. This is more reason to state the limitation of the
definition/data back on line 121-123.

175    Good point. We've addressed this by adding a clear statement of the limitation in line 136-137.

Line 265: OK, here the LLJ limitation is acknowledged. I think it would be helpful to mention something about this
back on lines 121-123 staring that more will be said about it later.

Yes, it's a good point. Modification has been made to line 136-137.

---

## Author Comment (AC2)

**Reviewer #2**

This study uses Self-Organizing Maps (SOM) and K-means clustering to reduce data dimensionality and identify the key components of variables such as Z500, which describe different large-scale meteorological patterns (LSMPs) that produce varying wind patterns. For each LSMP, wind data from HRRR is evaluated against two in-situ lidar buoys,

5 with biases documented and presented. I found the manuscript technically sound, with interesting and valid methods to present the results. Below are several suggestions that could help improve the motivation and discussion sections:

We thank the reviewer for their thoughtful and constructive comments and suggestions, which has substantially improved the quality of the manuscript. We have addressed all the reviewer's concerns and revised the manuscript accordingly. Our point-by-point responses are in blue and the modifications to the manuscript are quoted in green.

10 In the introduction, you mentioned that the community relies on modeling data, but you didn't explain why. A transition paragraph discussing the sparsity of observational data offshore is needed. This would provide context as to why models are essential and why it's important to validate them.

Good point. The following paragraph has been added to Line 32-37 to discuss the sparsity of observational data.

As of October 2023, five wind energy lease areas were established off the California coast — two off Humboldt County and
three off Morro Bay (BOEM, 2023). Observational datasets are ideal for assessing and characterizing the wind resource. The U.S. Department of Energy funded the installation of two research buoys in these areas, equipped with lidar and other instruments to collect wind measurements for resource assessment and model evaluation (Krishnamurthy et al., 2023). However, due to the challenges associated with deploying and maintaining offshore equipment, these measurements remain limited.

20 Although your focus is on validating HRRR, it might be worthwhile to mention other modeling datasets, especially since HRRR has a relatively short record. For instance, you could reference NOW-23 (offshore wind data developed by NREL) or the newly published Wind Toolkit (WTK-LED) by NREL. These datasets could also serve wind resource assessment purposes.

The following sentences have been included in the revision in Line 41-43.

25 The 2023 National Offshore Wind dataset (NOW-23) is the latest wind resource dataset for the offshore region in the U.S., launched by the National Renewable Energy Laboratory (NREL) (Bodini et al., 2024). The NOW-23 dataset delivers an updated and cutting-edge product to stakeholders.

The method you developed for validating HRRR model performance under different LSMPs can be applied to other model products as well. It might be helpful to mention this in the discussion to highlight the broader applicability of your approach.

Thank you for your insightful suggestion. We agree that the method developed for validating the HRRR model's performance under different LSMPs can be extended to other model products. Additionally, this approach can be applied to understand how environmental factors influence airflow evolution, benefiting predictive studies. By linking model performance with LSMPs, the results can also foster mechanism analysis, helping to explore the model's ability

35 to capture underlying physical processes. We have included a discussion highlighting the broader applicability of this method in the revised manuscript in Line 331-.

This study introduces a new approach to characterizing offshore winds and associated model biases, linking them to LSMPs. The methodology used for evaluating HRRR performance under different LSMPs can be applied to other numerical weather prediction models. This approach not only identifies model strengths and weaknesses but also provides valuable insights into

40 how environmental factors influence airflow, aiding predictive studies. By connecting model performance to LSMPs, this method promotes mechanism analysis, fostering studies on a deeper understanding of the physical processes behind wind patterns. Furthermore, the results are anticipated to guide the selection of cases for studying the influence of specific large-

30

scale and local factors on winds off the California coast, which will aid in refining numerical weather prediction models, thereby enhancing the efficiency and reliability of offshore wind energy production.

45 Finally, while this paper proposes a useful method for validating models beyond just examining overall mean wind speeds, it would be valuable to discuss the implications for industry and data users. How should they interpret the identified biases when using these data for wind farm development? What practical guidance can be offered?

In the revised manuscript, we expand on how the identified biases can inform wind farm development and offer practical guidance on using model outputs. The following texts has been added.

- 50 The identified model biases have significant implications for wind farm development, particularly in offshore environments where accurate wind resource assessments are essential. For instance, the overestimation of wind speeds in certain LSMPs, such as pre-ridge and California-high conditions, could result in overestimating potential energy output. To address this, data users should approach HRRR model outputs cautiously under these conditions and incorporate model uncertainties into their assessments. Beyond the mean status of wind speed, future studies could link the wind power features like ramp frequency
- 55 and intensity to LSMPs. Practical measures, such as utilizing ensemble forecasts or combining multiple models, can help mitigate the effects of these biases on wind farm siting and design decisions.

---

## Author Response (AR2)

We thank the reviewer for their thoughtful and constructive comments and suggestions, which has substantially improved the quality of the manuscript. We have addressed all the reviewer's concerns and revised the manuscript accordingly. Our point-by-point responses are in blue and the modifications to the manuscript are quoted in green.

5   **Reviewer #1**

In my first review, I had two main concerns about the paper. The first of these had to do with the superiority of the two-step process to reduce the number of SOMs over simply using a smaller number of SOMs to begin with. The authors have kindly provided this comparison in Fig. R3 of their response. I see two distinct differences between the two methods. As the author's point out, the two-step procedure in the top row has in the last panel a region of high gpm in the center-
10   west portion of the domain which is not present in any of the lower set of panels for the straight 5-SOM method. However, the lower set of panels also has in panel 2 a pattern of low GPM in the southwest portion of the domain, which is not present in any of the panels in the top row. Which of the two is better? The author's claim that the top row is (subjectively) better, but that is not obvious to me. Is there a way to quantitatively measure this, through some metric of the greatest separation between the figure patterns? The authors also have decided not to include Fig. R3 in the paper, but to me it is
15   an interesting result, worthy of inclusion and discussion. Without it, the reader is forced to simply accept the authors contention that the 2-step method is better, without being able to understand or appreciate the differences between the two.

We thank the reviewer for their detailed observations regarding the comparison between the two-step SOM/K-means method and directly applying a 5-SOM clustering approach. To address this, we provide a response from both physical
20   and statistical perspectives:

Physically, the 10x10 SOM step (Figure 1 in the manuscript) generates a lower-dimensional representation of the full dataset, capturing finer-scale details of the large-scale meteorological patterns. For instance, the Fig R3 second panel's pattern (low geopotential height anomaly in the southwest) emerges clearly in the upper-right portion of the 10x10 SOM map. This pattern is subsequently aggregated into a land-ocean pattern during the second (K-means) clustering step.
25   However, the high geopotential height anomaly over the central and western portion of the domain (representing the California high) observed in the 10x10 SOM prototypes is not captured when directly applying the 5-SOM clustering method. This demonstrates that the two-step approach preserves important physical features that the 5-SOM method misses, which are meteorologically significant for understanding regional wind variability.

Statistical, we calculated the Silhouette coefficient, a metric that measures the compactness of clusters and their
30   separation. A higher silhouette coefficient indicates that clusters are well-defined and distinct. As shown in the figure below:

[Figure]

**Figure Silhouette coefficients for each clusters in (a) two-state method and (b) 5-SOM clustering. The red dashed line indicates the mean Silhouette score.**

35

The two-stage clustering approach produces clusters with higher average silhouette coefficients (0.12) compared to the direct 5-SOM clustering (0.09). The improved silhouette scores indicate that the two-stage approach results in more distinct and well-separated clusters, confirming its statistical superiority over the 5-SOM method.

Since the physical perspective of the two-stage clustering has already been documented in Section 2.4 of the manuscript,
40  we have added the following text to **Lines 124–126** to provide a quantitative comparison:

We also compared the results of direct 5-SOM clustering with our two-stage method. The average silhouette coefficient is 0.12 for the two-stage method, compared to 0.09 for the direct 5-SOM clustering. The larger SS indicates that the resulting clusters are more well-defined and distinct.

45  The second main concern had to do with the "outlier" windspeeds present in the individual LSMP clusters shown in Fig. 4. The author's response to my concern is to note that in fact very different wind speeds can occur within a given cluster. That is in itself an important aspect of the method that should be emphasized more in the text. Also, it contradicts the statement in the abstract that "Distinct wind speed, wind direction, diurnal variation, and jet feature responses are observed for each LSMP and at both lidar locations." Fig. 4 shows that distinct wind speeds are not associated with each
50  LMSP. Instead, within 8 of the 10 outlined LSMPs shown in the figure, the wind speed varies from ~4 to ~14 m/s. Also, multiple LSMPs appear by eye to have almost identical distributions of wind speeds. Again, I think that the paper would be stronger if these issues were discussed in the manuscript. No method is perfect, and the science community would benefit more if some of the limitations of the SOM/K-means method were acknowledged and addressed.

We appreciate the reviewer's insightful feedback regarding the variability of wind speeds within individual LSMP
55  clusters. We agree that it is important to emphasize that while the LSMP-based classification method successfully links large-scale meteorological patterns to observed offshore winds, it does not guarantee a perfect one-to-one correspondence between an LSMP and a single "characteristic" wind speed. Indeed, different SOM prototypes within the same LSMP cluster may display a broad range of wind speeds, as noted by the reviewer.

This variability arises because offshore wind conditions result from the complex interplay between large-scale
60  circulations and localized factors such as coastal topography, sea surface temperature gradients, or mesoscale phenomena.

Even within a single LSMP, subtle differences in weather features can lead to substantial variation in wind speed. Thus, while the LSMP clustering serves as a useful framework to understand and categorize overarching synoptic patterns, it does not eliminate the inherent variability present in real-world wind data.

In the revision, we have changed the sentence in the abstract mentioned by the reviewer to (**Lines 16-18**):

While each LSMP is associated with characteristic large-scale atmospheric conditions and corresponding differences in wind direction, diurnal variation, and jet features at the two lidar sites, substantial variability in wind speeds can still occur within each LSMP

In the Results section, we added the following paragraph (**Lines 238-246**):

It is important to note that the classification of LSMPs does not imply that each LSMP is associated with a narrowly defined wind speed. Rather, each LSMP reflects a dominant synoptic environment under which the prevailing direction and magnitude of offshore winds are modulated. Within any single LSMP, a range of mesoscale and local factors (e.g., frontal passages, varying thermal contrasts, topographic influences, and boundary layer structures) can lead to substantial variability in observed wind speeds. For instance, under post-ridge conditions, most SOM prototypes show a decrease in wind speed due to the induced southerly anomaly; however, a few prototypes exhibit unexpectedly high wind speeds when the southerly anomaly surpasses the prevailing northerly flow. This illustrates that while the LSMP framework provides a useful synoptic-scale context, it is primarily a classification tool rather than a deterministic method, and thus cannot eliminate the inherent complexity and spread in the local wind speed distributions.

And in the Conclusion section, we added (**Lines 318-320**):

Nonetheless, these results should not be interpreted to mean that each LSMP strictly enforces a single wind speed regime. A wide range of wind speed outcomes can occur, influenced by local and mesoscale processes.